# Evaluation of Selected Amateur Rain Gauges with Hellmann Rain Gauge Measurements

**Radosław Droździoł** [1] and **Damian Absalon** [2,*]

1 Institute of Meteorology and Water Management, National Research Institute, Podleśna 61, 01-673 Warsaw, Poland; radoslaw.drozdziol@imgw.pl
2 Institute of Earth Sciences, Faculty of Natural Sciences, University of Silesia in Katowice, Będzińska 60, 41-200 Sosnowiec, Poland
* Correspondence: damian.absalon@us.edu.pl

**Abstract:** The paper compares precipitation measurements from the Stratus manual rain gauge from the CoCoRaHS network and two Davis Vantage Vue and Davis Vantage Pro 2A rain gauges with the Hellmann rain gauge. Comparative measurements were made on a specially prepared experimental plot. The statistical calculations took into account 15 full months in the period from 1 October 2019, to 31 December 2020. In order to estimate the differences in measurements between amateur rain gauges and the Hellmann rain gauge, two statistics were calculated: the mean square error (MSE) and the mean absolute error (MAE). In order to present formal and more detailed differences in measurements between amateur rain gauges and the Hellmann rain gauge, analyses were performed using a linear regression model. The general form of the tested models was presented. The procedure for estimating the parameters of the models and the method of comparing the fit of the models to the data were described, and the rain gauge whose measurements were most closely related to the measurements of the Hellmann rain gauge was indicated. The study showed that the higher price of amateur rain gauges does not mean higher quality. The study showed that the Stratus rain gauge was the best at recording daily precipitation totals. The Davis Vantage Pro 2A rain gauge recorded daily precipitation sums with less accuracy. The Davis Vantage Vue rain gauge, despite being located on the roof, recorded similar rainfall totals as the Hellmann rain gauge. It was found that, despite the different construction and measurement methods, the precipitation measurement data from the Stratus rain gauge and the Davis Vantage Vue rain gauge are suitable both for climate monitoring and for use, after applying quality control, in NMHS networks.

**Keywords:** personal weather stations; PWS; meteorological measurements; comparison of measurements; precipitation; CoCoRaHS; Davis Vantage; Hellmann

## 1. Introduction

Precipitation that reaches the Earth's surface is defined as the liquid or solid product of water vapor condensation falling from clouds or deposited from the air to the ground. Precipitation includes the following forms: rain, drizzle, hail, snow, grit, dew, frost, rime and fog. Precipitation measurements are made in order to obtain the most accurate information possible about the amount of precipitation occurring at a specific time and place. Quantitative and qualitative understanding of precipitation is of great importance in many fields. In synoptic meteorology, it is important to measure the total amount of precipitation. In hydrology, apart from the amount of precipitation, the possibility of obtaining information on the intensity of precipitation is also of great importance. Information on the timing of precipitation is crucial for agriculture, communication, supply, water management, hydropower, and public information. Precipitation is a special parameter in meteorology; therefore, it should be measured with high accuracy. However, its measurement, contrary to appearances, is difficult. Point precipitation measurements are not very representative

due to the high spatial and temporal variability of precipitation phenomena. Point measurement results can be easily affected by the conditions around the measurement site. The result of point measurement also depends on the type of exposure of the measuring instruments. Eventually, point measurements of rainfall, especially with the use of various types of rain gauges, are burdened with many methodological systematic errors of variable value, which together can completely distort the measurement result.

Personal weather stations (PWS) are devices used by enthusiasts of meteorological measurements to monitor the state of the atmosphere. Easily available, for a relatively low price, they have become a popular hobby almost worldwide. From year to year, the number of PWS systematically increases. PWS users can easily share their measurements in real time on various Internet platforms. Popular Internet portals, such as Weather Underground, Weathercloud, PWS Weather, AWEKAS, and CWOP, collect measurements from PWS users and offer access to visualized data in real time or at various time intervals, usually from 5 to 10 min. The number of PWS varies by continent and country. There are approximately 1700 devices in the UK [1]. In Poland, depending on the platform, their number was different (Figure 1), and in 2020, it ranged from 32 in the AWEKAS network to 1054 in the Weather Underground network. In the network of the Institute of Meteorology and Water Management—National Research Institute (IMGW-PIB) in Poland, in 2020, there were 500 telemetry stations.

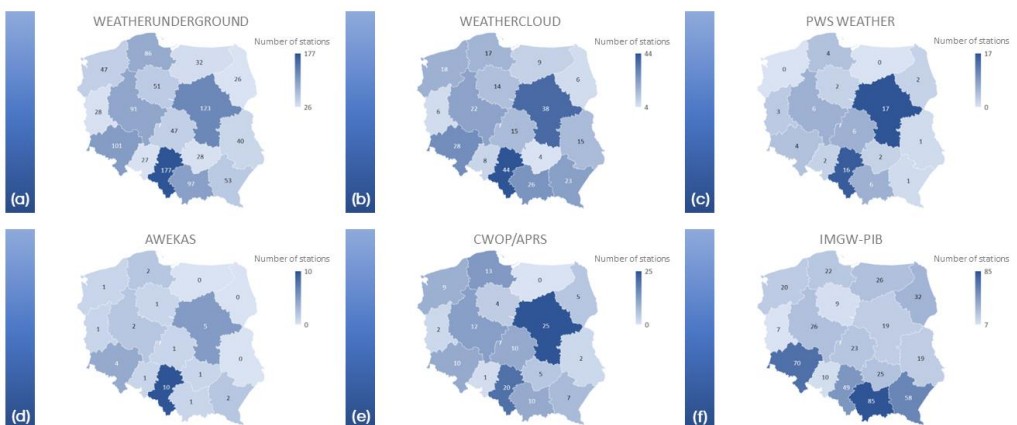

**Figure 1.** The number of PWS in Poland in individual voivodships in various Internet portals (**a**–**e**) and the number of stations in the IMGW-PIB network (**f**).

Most PWS are equipped with automatic rain gauges, thanks to which parameters, such as the amount of precipitation, the intensity of precipitation, and the time of precipitation start, end, and duration, are measured. This source of new data has become an opportunity for many National Meteorological and Hydrological Services (NMHS) to densify their operational rain gauge networks. The expansion of the current observation and measurement networks is associated with huge purchase and maintenance costs. Rainfall measurements from PWS have also become a challenge for NMHS. Therefore, in the last decade, the research concerned all parameters measured by PWS [1–3], a single parameter to be used in the analysis of urban climate and urban heat island [4–8], and precipitation [9–11].

In addition to automatic amateur rain gauges, manual rain gauges have been very popular for a long time. In Poland, there is no portal linking the community conducting such rainfall measurements, but in the United States of America, since 1998, and later in Canada and in the Bahamas, there has been a large and constantly growing community of volunteers in the Community Collaborative Rain, Hail, and Snow Network (CoCoRaHS) [12]. CoCoRaHS brings together volunteers who measure precipitation using one type of rain gauge, which in Europe is known as the Stratus. This rain gauge meets the requirements of the National Weather Service (NWS) for measuring precipitation and is listed in the Catalogue of National Standard Precipitation Gauges of the World Meteorological Organization (WMO) [13].

In the literature, there are still not many studies and comparative measurements on experimental plots that would thoroughly analyze the real differences between the measurements of precipitation from personal rain gauges and NMHS measurements. Therefore, the aim of the present study was to determine the degree of credibility of the measurement results obtained from amateur rain gauges in relation to the results of the Hellmann rain gauge.

## 2. Materials and Methods

The aim of the study was to determine the degree of reliability of the measurement results obtained from two automatic rain gauges of amateur weather stations and one manual amateur rain gauge in relation to the results of the Hellmann rain gauge used on the network of the Institute of Meteorology and Water Management—National Research Institute (IMGW-PIB) in Poland and treated in this study as reference equipment. For this purpose, in Będzin (Figure 2), an experimental plot (Figure 3c) was created, where three rain gauges were installed: an automatic rain gauge of the Davis Vantage Pro 2A personal weather station, a manual amateur Stratus rain gauge, and a Hellmann rain gauge. In addition, an automatic rain gauge from the Davis Vantage Vue amateur weather station was installed on the roof of a nearby building, 63 m away from the experimental plot (Figure 3a,b).

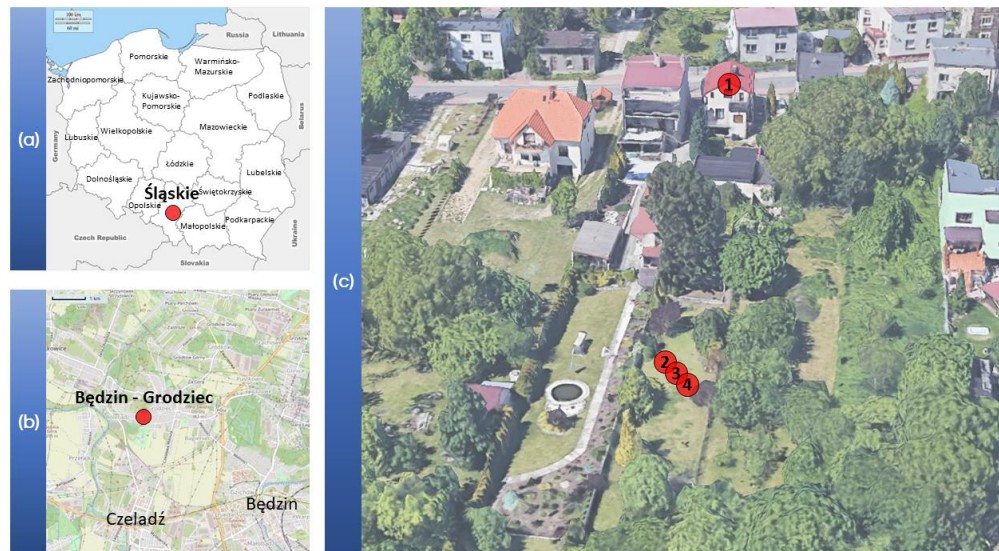

**Figure 2.** Location of rain gauges (**a**–**c**): (**c**) 1—Davis Vantage Vue rain gauge, (**c**) 2—Davis Vantage Pro 2A rain gauge, (**c**) 3—Hellmann rain gauge, (**c**) 4—Stratus rain gauge. Own study based on maps https://polska.geoportal2.pl/ (accessed on 15 April 2023) and https://www.google.pl/maps (accessed on 15 April 2023).

The Hellmann rain gauge was mounted 90 cm away from the Stratus rain gauge and 165 cm away from the Davis Vantage Pro 2A rain gauge. Throughout the entire measurement period, efforts were made to ensure that the surface of the experimental plot was as close to the natural state as possible. In the summer, the grass was mowed so that its height did not exceed 10–15 cm. In winter, the snow was left in its natural state until it disappeared, and the rain gauge posts were systematically cleared of snow. Throughout the year, the plot was kept in order, clearing it of all rubbish blown by the wind. Once a week, the rain gauges were inspected to prevent accumulation of dust, mud, cobwebs, etc. in the devices.

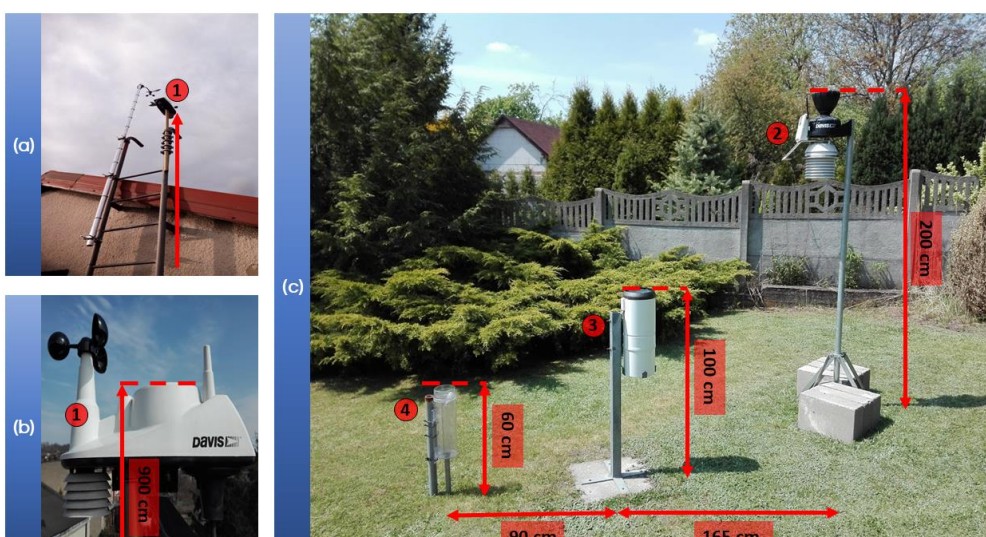

**Figure 3.** Location of rain gauges: (**a**,**b**) 1—Davis Vantage Vue rain gauge, (**c**) 2—Davis Vantage Pro 2A rain gauge, (**c**) 3—Hellmann rain gauge, (**c**) 4—Stratus rain gauge.

Precipitation measurements with manual rain gauges were carried out by the first author of the article, who had the necessary experience in carrying them out. Efforts were also made to ensure that the intervals between the readings from the Hellmann and Stratus rain gauges were as small as possible to prevent significant differences in readings. However, it should be noted that this may have influenced the results of the study.

The Hellmann rain gauge (Figure 4e) is used by IMGW-PIB at tier V stations (precipitation stations). These stations constitute the largest number of measurement points in the structure of the network of IMGW-PIB measurement stations [14]. The rain gauge inlet area was 200 cm$^2$ (Table 1). This device consisted of the following parts made of zinc sheet: base, receiver, tank, insert, and lid. The receiver of the rain gauge ended at the top with a sharp ring made of brass. The ring had a vertical slope on the inside and was cut diagonally on the outside. The bottom of the receiver was funnel-shaped with an open lower part that entered the neck of the water tank, set in a frame at the bottom of the base. The height of the entire rain gauge was set to 46 cm. During the period of snowfall, a special insert was placed in the receiver of the rain gauge in order to prevent the snow from being blown away. After this period, the insert was removed because its presence would be unacceptable, as during liquid precipitation it would increase the surface wetted by precipitation water, lowering the measurement results. The rain gauge was suspended from a holder attached to a metal post. According to the IMGW-PIB procedure, the height of the post protruding above the ground was 90 cm. The upper end of the post was sharply cut on the south side, which prevented snow from accumulating on it. The rain gauge holder was installed on the north side at such a height that the upper edge of the brass ring of the rain gauge tank was at a height of 10 cm above the sharply cut end of the post, i.e., above the ground at a height of 1 m (Figure 3c). The inlet surface of the installed rain gauge was exactly horizontal. The amount of precipitation was measured once a day, at 6:00 UTC, using a measuring tube (Figure 4f) with a scale, adjusted to 200 cm$^2$ of the rain gauge inlet area. On the scale, the intervals corresponding to the total millimeters of the precipitation height were separated by long lines, corresponding to 0.5 mm—medium lines, and corresponding to 0.1 mm—short lines. The measured precipitation was recorded in the observation diary with an accuracy of 0.1 mm. With a single pouring of water into the measuring tube, it was possible to measure a maximum of 10 mm of rainfall, the maximum rainfall in the rain gauge tank is 64 mm (Table 1), and in the case of the tank overflowing, a maximum of 192 mm of rainfall was contained in the base of the rain gauge.

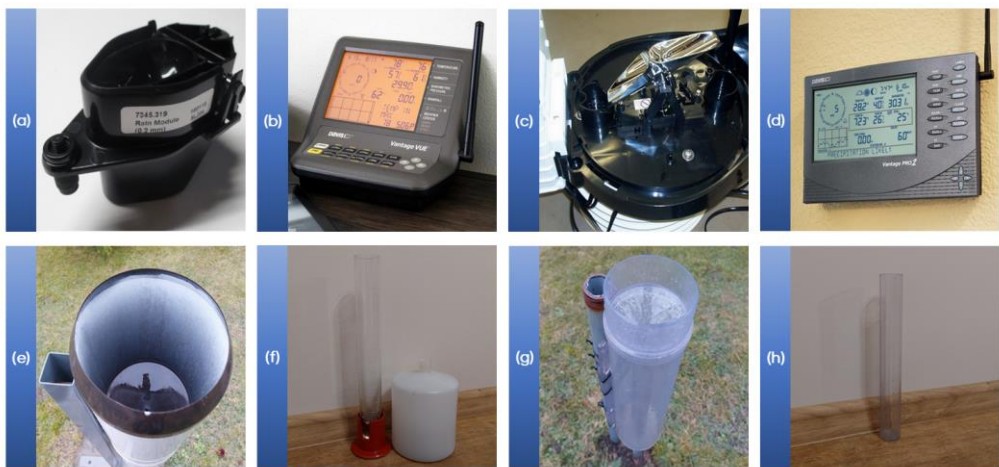

**Figure 4.** Automatic rain gauges with consoles: (**a**,**b**) Davis Vantage Vue, (**c**,**d**) Davis Vantage Pro 2A. Manual rain gauges with measuring cylinders: (**e**,**f**) Hellmann, (**g**,**h**) Stratus.

**Table 1.** Specification of rain gauges.

| Station Number | Device Type | Rain Gauge Measurement Method | Rain Gauge Inlet Area | Daily Measurement Range | Measurement Resolution | Measurement Accuracy | Price |
|---|---|---|---|---|---|---|---|
| 1 | Davis Vantage Vue | tipping bucket (automatic) | **116 cm²** | 0–999.7 mm | **0.2 mm** | 4% | 424.99 $ |
| 2 | Davis Vantage Pro2 A | tipping bucket (automatic) | **214 cm²** | 0–999.8 mm | **0.2 mm** | 3% | 1064.65 $ |
| 3 | Hellmann | manual | **200 cm²** | **0–64 mm** **0–192 mm** | **0.1 mm** | observer dependent | 769.21 $ |
| 4 | Stratus | manual | **78.5 cm²** | **0–25 mm** **0–266 mm** | **0.2 mm (possible 0.1 mm)** | observer dependent | 40.99 $ |

Values measured by the authors are shown in bold. The other values are taken from the manuals of the measuring instruments.

According to research carried out under the auspices of the WMO, where the Hellmann rain gauge was tested on several testing sites for a long time, it was found [15] that in certain conditions of stronger wind systematic wind errors occur in the data. Despite this, it is currently the most widely used rain gauge in the world and the WMO-accepted standard instrument. It is used in meteorological services in over 30 countries, including the Polish IMGW-PIB. The number of these instruments in national networks exceeds 30,000, and it is also used in Poland to perform comparative measurements with automatic professional and amateur rain gauges. With regard to the expectations of operational hydrology, which requires the provision of accurate values of area precipitation and forecasts of this precipitation, methods integrating precipitation data from Hellmann rain gauges and automatic rain gauges with data from a radar and satellite system are increasingly used [16,17]. However, in this work, they were not used due to the small distances between individual rain gauges.

The Stratus amateur manual rain gauge (Figure 4g) is a design used by amateur observers around the world. It is also the rain gauge used by the Community Collaborative Rain, Hail, and Snow Network (CoCoRaHS) [18]. The device consists of the following parts, made of high-quality plastic: base, receiver, a graduated tank, which is also a measure for measuring the height of precipitation, and a holder. The rain gauge receiver has a sharp plastic tip at the top and the inlet area is 78.5 cm² (Table 1). From the inside, the end of the receiver has a vertical fall, and from the outside, it is beveled. The bottom of the receiver is made in the shape of a funnel, the lower part of which is open and enters the measure, which is set in the frame at the bottom of the base. The height of the assembled rain gauge is 35 cm. During the period of snowfall, the receiver and the measuring cup were removed

from the rain gauge. The rain gauge was suspended on a holder attached to a 54 cm high metal post. The holder was attached to the post on the north side and the top edge of the plastic ring of the rain gauge was 4 cm above the end of the post, i.e., 60 cm above the ground (Figure 3c). The surface of the inlet was exactly horizontal, which was checked with a spirit level when installing the rain gauge. When performing the measurement, attention was paid to ensure that the graduated tank was properly positioned in the frame at the bottom of the base, and that the base was well placed on the holder. The amount of precipitation captured by the rain gauge was measured once a day at 6:00 UTC only with the measure (Figure 4h), which was included in the entire set. The measuring tube was equipped with a scale adapted to the inlet surface of the rain gauge. On the scale, the intervals corresponding to the total millimeters of the precipitation height were separated by long lines, and those corresponding to 0.2 mm by short lines. The measured precipitation was recorded in the observation diary with an accuracy of 0.1 mm, despite the 0.2 mm scale. The maximum rainfall contained in the measuring cylinder is 25 mm (Table 1), and in the case of overflowing the cylinder, the base of the rain gauge contained a maximum of 266 mm of rainfall.

The Davis Vantage Vue and Davis Vantage Pro 2A automatic amateur rain gauges are recording devices intended only for measuring and recording the amount of liquid precipitation (the rain gauges did not have the heating function).

The Davis Vantage Vue station is a set of integrated sensors, which includes an air temperature sensor, air humidity sensor, anemometer, and a rain gauge. The entire set was mounted on a metal tube, 1.5 m above the roof surface (Figure 3a), in 2014. The top edge of the plastic rain gauge ring was 900 cm above the ground (Figure 3b). The receiver of the rain gauge ended at the top with a sharp plastic tip. The inlet area was 116 cm$^2$ (Table 1). Both from the inside and outside the receiver was cut diagonally. The bottom of the receiver was made in the shape of a funnel, the lower part of which was open, secured with a clip for catching dirt. Under the opening, there was a single-chamber tipping bucket (Figure 4a), measuring 0.2 mm of precipitation (Table 1). A magnet was attached to the bucket, that moved towards a magnetic switch (reed switch). As a result of the impact of the magnetic field, the electrodes of the measuring switch shorted, closing the electric circuit. The short circuit was recorded by the recorder. Impulse data was sent wirelessly to the rainfall recording console (Figure 5b). The console was equipped with a data logger and the data recording interval was 5 min. In this configuration, the recorder, not connected to a computer, could save 2560 records—8 days of measurements. Throughout the measurement period, the data logger console was connected via a USB cable to the computer, where, using the Cumulus 2 program (Figure 5c), the data was saved every 5 min in the form of a text file (Figure 5d). The set of sensors of the Davis Vantage Vue station was powered by a solar panel, storage battery, and a CR-123 battery, and the daily measuring range of liquid precipitation, according to the manufacturer, was a maximum of 999.7 mm [19].

The Davis Vantage Pro 2A is a very popular amateur station in Poland and around the world. This model is often used in agriculture, industry, environmental protection, teaching, and research. The station included a set of integrated sensors: air temperature, air humidity, and a rain gauge. The anemometer could be separated from the station and measure at a different altitude. The rain gauge was mounted on a metal tube 188 cm above the ground. The top edge of the rain gauge was at a height of 200 cm (Figure 3c). The rain gauge consisted of two plastic parts: a receiver and a base (Figure 4c). The rain gauge receiver had a sharp tip at the top, next to which metal pins were mounted on the outer part to prevent birds that generated dirt (feathers, droppings) from sitting on it. The inlet area was 214 cm$^2$ (Table 1). The bottom of the receiver was made in the shape of a funnel, the lower part of which was open, secured with a removable strainer for collecting dirt. Under the opening, there was a two-chamber tipping bucket (Figure 4c), measuring 0.2 mm of precipitation (Table 1). The construction of the tipping bucket, located in the base of the rain gauge, was based on the principle of balance scales. The device consisted of two twin, open metal chambers resting on a common bearing. Above the chambers (in

the vertical axis of symmetry), there was a funnel outlet of the rain gauge receiver. The empty bucket had two rest positions; tilted to the left or right to the stop, in which each of the chambers was alternately placed under the funnel outlet. Since the center of gravity of each filled chamber exceeded the support point, a torque was generated, and the trough capsized as soon as it was loaded with a certain amount of precipitation (0.2 mm). Water from the chamber was poured into the drainage funnels of the rain gauge base, from where it was poured into the environment. At the same time, the other chamber was placed under the funnel's mouth. During each tipping of the tray in the associated electrical circuit, an impulse with appropriate parameters was triggered. The sum of the pulses counted during the precipitation event enabled the calculation of the precipitation height, which was related to the duration of the precipitation intensity. The data was then wirelessly sent via a retransmitter (Figure 5f) to the rainfall recording console (Figure 5g). The console was equipped with a data logger with a data recording interval of 5 min. The console with the data logger was connected via a USB cable to the computer, where, through the Cumulus MX program (Figure 5h), the data was saved in the form of a text file. The set of sensors was powered by a solar panel, a storage battery, and a CR-123 battery. The daily range of liquid precipitation measurement according to the manufacturer was a maximum of 999.8 mm [19].

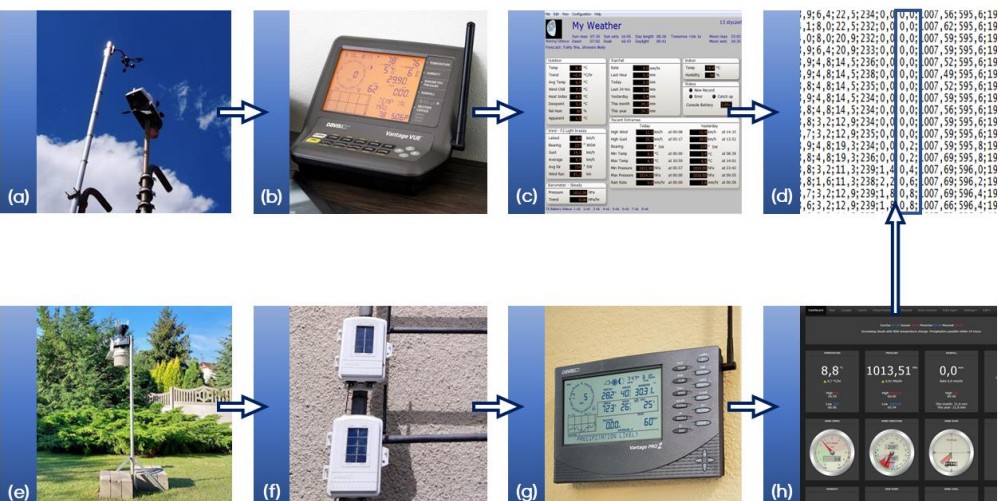

**Figure 5.** Data transfer scheme: (**a**) Davis Vantage Vue station, (**b**) data receiving console, (**c**) Cumulus 2 software for data presentation and recording, (**d**) data recording scheme, (**e**) Davis Vantage Pro 2A station, (**f**) retransmitter, (**g**) data receiving console, and (**h**) Cumulus MX program.

Comparative measurements were carried out from 1 October 2019, to 31 December 2020, on the experimental plot in Będzin Grodziec. The daily rainfall totals of the Hellmann and Stratus rain gauges were totals between 6:00 UTC the previous day and 6:00 UTC the current day. The software of the Davis Vantage Vue and Davis Vantage Pro 2A automatic stations recorded data at 5-min intervals and summed up the amount of daily precipitation that occurred between 0:00 and 24:00. The daily sums from the automatic stations were recalculated so that the measurements covered the period as in the case of the daily sums of manual rain gauges. In total, 461 daily measurements were analyzed.

Statistical analyses were performed using the statistical package R 4.0.2. Raw data are presented in graphs (Figure 6), while descriptive statistics and basic relationships between measurements from amateur instruments and the measurement from the Hellmann rain gauge are presented in the table (Table 2).

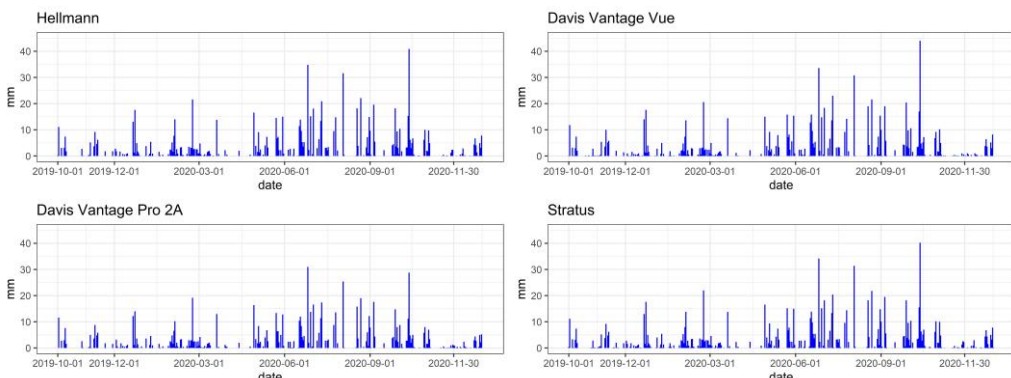

**Figure 6.** Time distribution of atmospheric measurements from the analyzed rain gauges.

**Table 2.** Descriptive statistics for precipitation measurements.

| Rain Gauge | M | SD | Min | $Q_1$ | Me | $Q_3$ | Max | P (zero) | MSE | MAE | ρ |
|---|---|---|---|---|---|---|---|---|---|---|---|
| Hellmann | 4.45 | 6.24 | 0.1 | 0.5 | 2.3 | 5.20 | 40.9 | 0.55 | | | |
| Davis VantageVue | 4.39 | 6.40 | 0.2 | 0.4 | 2.0 | 5.20 | 44.0 | 0.54 | 0.27 | 0.22 | 0.90 |
| Davis Vantage Pro 2A | 3.91 | 5.24 | 0.2 | 0.6 | 2.1 | 4.65 | 31.0 | 0.55 | 1.00 | 0.37 | 0.89 |
| Stratus | 4.21 | 6.01 | 0.1 | 0.5 | 2.1 | 4.80 | 40.2 | 0.51 | 0.02 | 0.08 | 0.97 |

M—mean, SD—standard deviation, Min—minimum, Q1—first quartile, Me—median (second quartile), Q3—third quartile, Max—maximum, P (zero)—proportion of rainless days, MSE—mean square error, MAE—mean absolute error, ρ—Spearman's ρ correlation coefficient. The error in MSE and MAE referred to the difference between the Hellmann measurement and the measurement from a given amateur instrument.

## 3. Results and Discussion

At the descriptive level, the measurements from the Davis Vantage Pro 2A rain gauge differed the most from the measurements from the Hellmann rain gauge, especially for the mean and standard deviation of precipitation. The values of Spearman's non-parametric correlation coefficient, ρ, indicated that the measurements from the Stratus rain gauge were the most closely related to the measurements from the Hellmann rain gauge.

In order to initially estimate the differences in measurements between amateur rain gauges and the Hellmann rain gauge, two statistics were calculated: the mean square error (MSE) and the mean absolute error (MAE), defined as:

$$\text{MSE} = \frac{1}{n} \sum_{i=1}^{n} (y_i - x_i)^2, \tag{1}$$

$$\text{MAE} = \frac{1}{n} \sum_{i=1}^{n} |y_i - x_i|, \tag{2}$$

where n is the number of measurements, $y_i$ is the i-th measurement from the Hellmann rain gauge, and $x_i$ is the i-th measurement from the amateur rain gauge. Both statistics showed that the Stratus rain gauge was characterized by the smallest error of measurement, compared to the measurement from the Hellmann rain gauge.

### 3.1. Modeling

In order to present the differences in measurements between amateur rain gauges and the Hellmann rain gauge formally and in more detail, analyses were performed using thr linear regression model in the next step. Section 3.1.1 presents the general form of the tested models. Section 3.1.2 briefly describes the procedure for estimating model parameters and the method for comparing the fit of the models to the data. Thanks to the comparison of models in which the predictors are measurements from amateur rain gauges, in Section 3.1.3, the rain gauge whose measurements are most strongly related

to the measurements with the Hellmann rain gauge is indicated and the dependence of measurement errors on selected measurement periods is tested.

### 3.1.1. Description of the Models

The relationship between the measurement from the Hellmann rain gauge Y and the measurement from the amateur rain gauge X can be described using the linear regression equation:

$$y_i = \beta_0 + \beta_1 x_i + \epsilon_i,$$
$$\epsilon_i \sim N(0, \sigma), \tag{3}$$

where $\beta_0$ is a constant that, for the centered predictor $x_i$ (that is, after subtracting the mean value of X from each $x_i$ value), represents the expected mean value of Y for the mean value of X. The term $\beta_1$ is the strength of the relationship between Y and X and the predicted change in the y value, after changing the value of x by one unit (i.e., by 1 mm). Finally, the parameter $\sigma$ is the standard deviation of the prediction error, $\epsilon$, which in this application can be interpreted as the average prediction error of the Y values predicted from the X values. Smaller values of $\sigma$ indicate that the X rain gauge measurements are closer to the Hellmann rain gauge measurements.

In order to test whether the mean measurement error can be dependent on the measurement period, G, the model in Equation (3) can be extended with the assumption that the prediction error depends on the measurement period, G:

$$\log(\epsilon_i) = \sigma_0 + \sigma_1 G_i, \tag{4}$$

where $G_i$ is a matrix of orthogonal contrasts (with mean 0) coding the measurement period to which the i-th observation belongs, $\sigma_0$ is a parameter representing the average prediction error, and $\sigma_1$ is a vector of parameters (one for each column in G), whose values (simplified) represent the differences between the average prediction error, $\sigma_0$, and a given measurement period.

Finally, it is also possible that the measurement error depends on the amount of precipitation on a given day. In order to test whether the measurement error can change proportionally to the amount of precipitation, the model from Equation (3) can be supplemented with the following component:

$$\log(\epsilon_i) = \sigma_0 + \sigma_1 x_i. \tag{5}$$

In model 5, the interpretation of $\sigma_0$ is analogous to model 4 (assuming that the mean $x_i$ equals zero), while $\sigma_1$ represents the predicted change in the prediction error on a logarithmic scale for a change of one unit in $x_i$.

### 3.1.2. Estimation and Comparison of Models

Estimation of model parameters was performed in Bayesian terms using the brms package [20,21]. In the Bayesian analysis, the a posteriori probability distribution of the model parameters (e.g., prediction error $\sigma$) is subject to estimation, calculated using likelihood integration and a priori distribution. In these analyses, poorly informative a priori and data-justified distributions were used [22]. Inference about the statistical reliability of a parameter (e.g., difference between measurements) is made by calculating the mean and 95% credible interval (CI). A statistically reliable effect is one for which the 95% CI is not zero [23,24]. In the brms package, the estimation of a posteriori distributions is performed using a sampling algorithm implemented in the STAN language [25]. The parameters of the sampling procedure were selected in a way that ensured a very good approximation of the a posteriori distribution.

Since each of the estimated models was fitted to the data of the same dependent variable Y (Hellmann rain gauge measurements), it is possible to compare the predictive power of both measurements from amateur stations and assumptions regarding the dependence of the prediction error on additional sensors. One of the best methods of comparing the

predictive power of models is the leave-one-out information criterion (LOOIC) statistic [26]. The LOOIC statistic is calculated based on the cross-validation procedure and represents the estimated predictive power of the model for new data. Since the LOOIC depends on the number of observations, the absolute values of the statistic have no direct interaction. In the context of model comparison, a difference in LOOIC values between two ΔLOOIC models is statistically significant when it exceeds twice the standard error of this SEΔLOOIC difference. Lower LOOIC values indicate a model better fitted to the data.

### 3.1.3. Comparison of the Predictive Power of Amateur Rain Gauges

Rainfall measurements of each of the amateur rain gauges were predictors for the Hellmann rain gauge measurements in five different models. The first model was the simplest model from Equation (3). The next three models were based on model 3 with an additional component from Equation (4), in which the season (S), month (M), or half-year (HY) was tested as the measurement period. The last model was model 3, with a component from Equation (5).

The results of the comparison, matching the data of each model, are presented in the table (Table 3). It was noted that for each of the amateur rain gauges, model 4 assumed differences in the average prediction error depending on the measurement period, which were no better fitted to the data than the simplest model. This means that for none of the amateur rain gauges, the prediction error was not dependent on any of the tested measurement periods. Model 5 turned out to be the best for each of the amateur rain gauges, which is indicated by statistically significantly lower LOOIC values compared to model 3.

**Table 3.** Comparison of predictive power of models.

| | Model | LOOIC | SE | ΔLOOIC | SE$_{\Delta LOOIC}$ | LOO R$^2$ |
|---|---|---|---|---|---|---|
| | 3 | 688.7 | 89.8 | 0.0 | 0.0 | |
| Davis Vantage Vue | 4 [S] | 716.2 | 114.4 | −27.5 | 45.3 | |
| | 4 [M] | 1163.9 | 457.6 | −475.3 | 431.8 | |
| | 4 [HY] | 700.2 | 92.5 | −11.5 | 4.5 | |
| | **5** | **583.0** | **115.9** | **105.6** | **49.9** | **0.989** |
| | 3 | 992.3 | 169.5 | 0.0 | 0.0 | |
| Davis Vantage Pro2 A | 4 [S] | 1073.9 | 221.9 | −81.5 | 67.9 | |
| | 4 [M] | 1351.3 | 451.3 | −359.0 | 403.6 | |
| | 4 [HY] | 974.1 | 160.6 | 18.2 | 43.9 | |
| | **5** | **675.2** | **119.3** | **317.2** | **169.4** | **0.977** |
| | 3 | −500.2 | 52.4 | 0.0 | 0.0 | |
| | 4 [S] | −496.8 | 54.7 | −3.4 | 11.1 | |
| Stratus | 4 [M] | −500.0 | 55.1 | −0.2 | 22.7 | |
| | 4 [HY] | −495.7 | 53.1 | −4.5 | 1.8 | |
| | **5** | **−639.2** | **53.4** | **139.0** | **42.6** | **0.998** |

ΔLOOIC—difference in LOOIC compared to model 3, SEΔLOOIC—standard error ΔLOOIC, LOO R2—R2 adjusted for LOOIC. Statistically significant differences are in bold.

The comparison of LOOIC between rain gauges clearly shows that the Stratus device had a much lower prediction error than Davis Vantage Vue, ΔLOOIC (SEΔLOOIC) = 1222.2 (125.4) and the Davis Vantage Pro2 A, ΔLOOIC (SEΔLOOIC) = 1314.4 (127.4).

### 3.1.4. Comparison of Parameters

The a posteriori distributions of the parameters of the best-fit models are shown in the graph (Figure 7). According to the results of the analysis based on LOOIC, the average values of the a posteriori distribution of the $\sigma_0$ parameter for the Stratus rain gauge, $\sigma_0 = |0.1|$, 95% CI: [0.07, 0.13], were significantly lower than for the other two amateur rain gauges. In addition, for this rain gauge, the parameter of dependence of the prediction error on precipitation $\sigma_1$ was the lowest, and the parameter of the strength of the relationship between the measurements $\beta_1$ was closest to the ideal value of 1 mm.

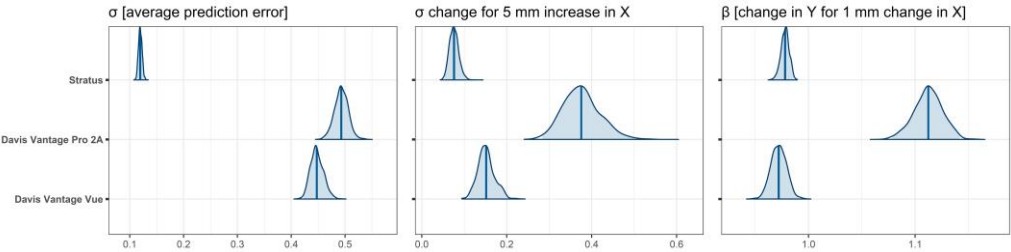

**Figure 7.** A posteriori distributions of the most important parameters of the best-fit models.

The relationships between amateur measurements and Hellmann rain gauge measurements, as well as predictions of the best-fit models, are shown in the figure (Figure 8).

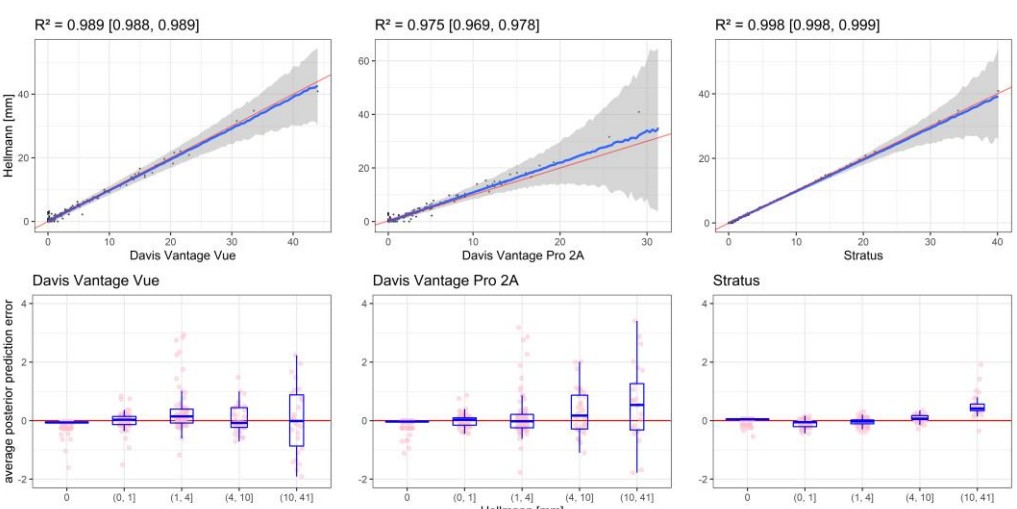

**Figure 8.** Predictions of measurements from the Hellmann rain gauge based on the best-fit models for each of the amateur rain gauges. The upper panels are scatterplots, where the grey point is one observation, the blue line is the linear component of the regression model, and the grey area is the 95% confidence interval of the model's prediction. The red line is the reference axis. The lower panels present the mean (from a posteriori distributions) prediction errors (pink points, in mm), grouped for selected rainfall ranges. Blue boxes represent standard boxplots. The red line shows a prediction error of zero.

Chart panels (Figure 8) clearly show a very high dependence of the prediction error of the Davis Vantage Pro 2A rain gauge on the amount of precipitation. Davis Vantage Vue and Davis Vantage Pro 2A recorded precipitation particularly badly when compared to the Hellmann rain gauge measured values in the range above 1 mm. For the relationship between the measurements from the Stratus and Hellmann rain gauge, a prediction with the smallest and relatively stable absolute error values was recorded. A significant increase in the prediction errors for the Stratus rain gauge was recorded only for the range above 10 mm of precipitation, as measured by the Hellmann rain gauge.

The largest daily difference in precipitation measured with the Stratus rain gauge compared to the Hellmann rain gauge was 0.7 mm. For the Davis Vantage Vue rain gauge,

it was 3.4 mm. For the Davis Vantage Pro 2A rain gauge, it was 12.1 mm. A comparison of differences in monthly precipitation sums (Table 4) showed similar relationships. The smallest differences from the Hellmann rain gauge were shown by the Stratus rain gauge, followed by the Davis Vantage Vue and the Davis Vantage Pro 2A. Comparing the sums of precipitation from the entire period of the measurement session, the situation was different. The Davis Vantage Vue rain gauge best recorded the precipitation total (−2.5 mm), followed by the Stratus (+30.0 mm) and Davis Vantage Pro 2A (−122.9 mm).

**Table 4.** Comparison of monthly precipitation totals (in mm).

| Month, Year | Hellmann Rain Gauge | Davis Vantage Vue Rain Gauge | Davis Vantage Pro 2A Rain Gauge | Stratus Rain Gauge |
|---|---|---|---|---|
| 10.2019 | 29.8 | 32.0 (+2.2) | 29.4 (−0.4) | 31.1 (+1.3) |
| 11.2019 | 33.2 | 34.2 (+1.0) | 32.2 (−1.0) | 35.6 (+2.4) |
| 12.2019 | 50.0 | 46.4 (−3.6) | 42.6 (−7.4) | 52.8 (+2.8) |
| 1.2020 | 19.2 | 17.2 (−2.0) | 18.4 (−0.8) | 23.0 (+3.8) |
| 2.2020 | 83.9 | 77.6 (−6.3) | 74.2 (−9.7) | 87.8 (+3.9) |
| 3.2020 | 31.2 | 29.2 (−2.0) | 29.2 (−2.0) | 32.5 (+1.3) |
| 4.2020 | 19.1 | 18.0 (−1.1) | 19.0 (−0.1) | 19.9 (+0.8) |
| 5.2020 | 82.6 | 91.0 (+8.4) | 78.8 (−3.8) | 86.9 (+4.3) |
| 6.2020 | 109.8 | 110.8 (+1.0) | 98.0 (−11.8) | 111.1 (+1.3) |
| 7.2020 | 101.1 | 100.6 (−0.5) | 87.6 (−13.5) | 101.9 (+0.8) |
| 8.2020 | 103.2 | 103.4 (+0.2) | 86.2 (−17.0) | 103.6 (+0.4) |
| 9.2020 | 77.7 | 80.0 (+2.3) | 65.8 (−11.9) | 79.3 (+1.6) |
| 10.2020 | 117.0 | 120.8 (+3.8) | 89.2 (−27.8) | 118.7 (+1.7) |
| 11.2020 | 25.3 | 22.0 (−3.3) | 18.4 (−6.9) | 27.2 (+1.9) |
| 12.2020 | 37.6 | 35.0 (−2.6) | 28.8 (−8.8) | 39.3 (+1.7) |
| 10–12.2019 | 113.0 | 112.6 (−0.4) | 104.2 (−8.8) | 119.5 (+6.5) |
| 1–12.2020 | 807.7 | 805.6 (−2.1) | 693.6 (−114.1) | 831.2 (+23.5) |
| total | 920.7 | 918.2 (−2.5) | 797.8 (−122.9) | 950.7 (+30.0) |

A comparison of rainy and rainless days (Figure 9) showed that precipitation was best recorded by the Stratus rain gauge. Every time the Hellmann rain gauge recorded precipitation, the Stratus rain gauge also recorded it. However, when Stratus showed rain, Hellmann showed rain only 91.6% of the time. Worse results were recorded for automatic rain gauges.

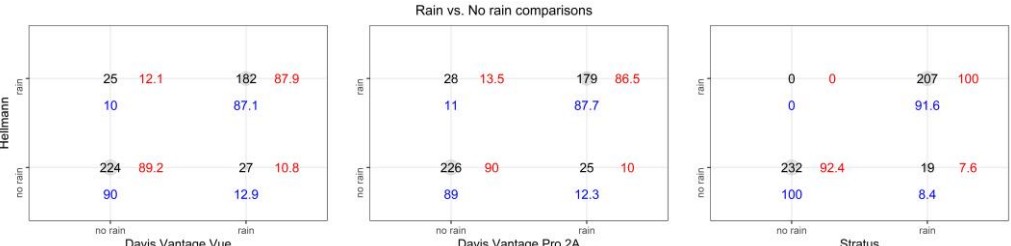

**Figure 9.** Comparison of rainy and rainless days in cross-tables. Black numbers are frequencies, red numbers are percentages that add up to 100 in rows, and blue numbers are percentages that add up to 100 in columns.

The authors concluded that the differences in measurements between manual rain gauges may result from their design and the lack of a wind shield, which is not used in



the Polish IMGW-PIB measurement and observation network. In addition, precipitation measured with the Stratus rain gauge was read with an accuracy of 0.1 mm, despite the 0.2 mm scale, and in the case of precipitation higher than 25 mm, the precipitation collected in the base of the rain gauge had to be poured into the measuring cup. These two cases were the cause of generating further, small errors, which eventually grew to a greater value of +30 mm in the entire measurement session (Table 4). In the case of automatic rain gauges, errors in measurements resulted from the design of the tipping bucket itself and the inability to perform measurements of 0.1 mm. The bucket tilted and recorded precipitation only at the value of 0.2 mm. It happened that precipitation of 0.1 mm remained in the tipping bucket until the next day, and only later, when precipitation occurred, was it recorded.

## 4. Conclusions

The study showed that the Stratus rain gauge was the best at recording daily precipitation totals. The Davis Vantage Pro 2A rain gauge recorded daily precipitation sums with less accuracy. The Davis Vantage Vue rain gauge, despite being located on the roof, recorded similar rainfall totals as the Hellmann rain gauge. The undoubted disadvantage of the Stratus rain gauge was the need for daily morning visits and manual measurement. Automatic rain gauges are definitely more convenient to use. The advantages of amateur automatic rain gauges with a tipping bucket include the possibility of almost maintenance-free operation, the possibility of performing rainfall measurements at very short intervals, the possibility of accurately determining the duration, start, and end of precipitation, and the simplicity of the instrument. One of the most important advantages of amateur automatic rain gauges was the almost immediate information about the occurrence of precipitation. The results of the present study confirm previous research results [27] that the measurements made with the Stratus rain gauge were of high quality. In most cases, this type of rain gauge worked comparably, and the difference in precipitation of the entire measurement session was +3.3% compared to the standard Hellmann rain gauge used at IMGW-PIB.

It is very difficult for automatic amateur tipping-bucket rain gauges to achieve the accuracy required by the WMO for precipitation measurements and requires the NMHS to apply quality control methods when using this type of data. The tests showed that the cumulative error during the entire measurement session was negative: −0.3% for the Davis Vantage Vue rain gauge and −13.3% for the Davis Vantage Pro 2A rain gauge, so depending on the type of rain gauge, the measured amount of precipitation will be slightly lower than the amount of precipitation measured with the Hellmann rain gauge. The small difference in the total precipitation of the entire period between the Davis Vantage Vue rain gauge, which was installed on the roof of the building, and the Hellmann rain gauge is noteworthy. Automatic amateur rain gauges, similar to those working in NMHS networks, do not distinguish the appearance of dew, frost, icing, fog condensate, etc., from atmospheric precipitation. Observations of this type, made by observers both in NMHS and amateur networks, are disappearing.

The results of precipitation measurements made with amateur rain gauges differed the most in relation to measurements made with a Hellmann rain gauge in the case of daily data. Depending on the amateur rain gauge, the differences blurred in the case of monthly data and the entire period of comparative measurements. From the statistical point of view, the analyzed series of data from measurements with the Hellmann rain gauge and amateur rain gauges show a strong correlation, but the comparison of measurements from individual measurement dates shows differences. In the case of the Stratus rain gauge, the recorded differences in the size of monthly precipitation totals were unidirectional. On the other hand, automatic amateur rain gauges indicated different precipitation sums and a smaller number of days with precipitation compared to the Hellmann rain gauge. The size of the differences and the degree of data consistency did not depend on the period of the cold and warm half-year in which the comparison was made. Despite the different construction and measurement methods, the precipitation data from the Stratus rain gauge

and the Davis Vantage Vue rain gauge are suitable both for climate monitoring and for use, after applying control, in NMHS networks. In order to ensure comparability and control of daily precipitation data, it is advisable to duplicate amateur sensors. The best solution is to have a manual rain gauge next to an automatic rain gauge. It is also necessary to calibrate the instruments and constantly verify the data, as well as further comparative studies, the result of which should be the introduction of correction factors for amateur rain gauges.

**Author Contributions:** Conceptualization, R.D.; methodology, R.D. and D.A.; software, R.D.; validation, R.D. and D.A.; formal analysis, R.D. and D.A.; investigation, R.D. and D.A.; resources, R.D. and D.A.; data curation, R.D. and D.A.; writing—original draft preparation, R.D.; writing—review and editing, R.D. and D.A.; visualization, R.D. All authors have read and agreed to the published version of the manuscript.

**Funding:** This research received no external funding.

**Data Availability Statement:** Please write an e-mail to the authors in order to receive data.

**Conflicts of Interest:** The authors declare no conflict of interest.

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
