# Peer review of "Evaluation of Selected Amateur Rain Gauges with Hellmann Rain Gauge Measurements"

_climate, doi:10.3390/cli11050107_

Round 1

Reviewer 1 Report

The paper compares precipitation measurements from the Stratus manual rain gauge from the CoCoRaHS network, two Davis Vantage Vue and Davis Vantage Pro 2A rain gauges with the 11 Hellmann rain gauge. Comparative measurements were made on a specially prepared experimental 12 plot. The statistical calculations took into account 15 full months in the period from October 1, 2019 13 to December 31, 2020. In order to estimate the differences in measurements between amateur rain 14 gauges and the Hellmann rain gauge, two statistics were calculated: the mean square error (MSE) 15 and the mean absolute error (MAE). In order to present formal and more detailed differences in 16 measurements between amateur rain gauges and the Hellmann rain gauge, analyzes were per-17 formed using a linear regression model. The general form of the tested models was presented. The procedure for estimating the parameters of the models and the method of comparing the fit of the 19 models to the data were described, and the rain gauge, whose measurements are most closely re-lated to the measurements with the Hellmann rain gauge, was indicated. The study showed that the 21 higher price of amateur rain gauges does not mean higher quality. The cheapest Stratus rain gauge recorded daily precipitation sums best, and the most expensive Davis Vantage Pro 2A the worst. It was found that, despite the different construction and measurement methods, the precipitation measurement data from the Stratus rain gauge and the Davis Vantage Vue rain gauge are suitable 25 both for climate monitoring and for use, after applying quality control, in NMHS networks.
This work is publishable in the journal after addressing following issues. I recommend to separate the section of discussion from the results. If it is not necessary, I recommend to remove the citations from the conclusion section. I recommend following studies to review and update your article with given studies (1-4)
(1)Evaluation and projection of precipitation in Pakistan using the Coupled Model Intercomparison Project Phase 6 model simulations. International Journal of Climatology
(2)Assessing the Impact of Long-Term ENSO, SST, and IOD Dynamics on Extreme Hydrological Events (EHEs) in the Kelani River Basin (KRB), Sri Lanka. Atmosphere 2023, 14, 79. https:// doi.org/10.3390/atmos14010079
(3)Statistical and qualitative evaluation of multi-sources for hydrological suitability in flood-prone areas. Journal of Hydrology. Volume 588
(4)Hydrological evaluation of satellite and reanalysis Precipitation Products in the glacier fed river basin (Gilgit). Arabian Journal of Geosciences
Overall, the article must have to update with latest studies having literature on the background of rain gauges, for reference (5) Precipitation measurements by manual and automatic rain gauges and their influence on homogeneity of long-term precipitation series
(6) Evaluation of precipitation measurements obtained from different types of rain gauges "

Overall, the article must have to update with latest studies.

Reviewer 2 Report

This manuscript compares daily precipitation measured by several different rain gauges at the same location (with one gauge is slightly away from the other three) for 15 months. Several statistics were calculated to evaluate amateur gauges compared with Hellmann gauge measurements. The authors conclude that “the higher price of amateur rain gauges does not mean higher quality”. While this study provides valuable insights to help readers better understand precipitation measurements, the conclusion seems overstated and lacks sufficient evidence to support it. Moreover, the analysis presented in this manuscript seems to lack depth in some areas. Therefore, it is imperative that the authors devote significant effort to improve the manuscript before it can be considered for publication.

Major comments:

To evaluate “amateur” rain gauges with treating Hellmann measurements as the truth, one should introduce why Hellmann gauge is more accurate what are the possible error sources for others. This is lacking in the current introduction.

Hellmann and Stratus rain gauges are manual gauges that reads once per day, while the two Davis gauges are automatic gauges with 5-min resolution. The study only compares the daily accumulated precipitation and concludes the expensive gauges (the Davis) perform worse than the cheaper one (Stratus). This is unfair as the Davis gauges are expensive because it is automatic and high-resolution. Moreover, the Davis Vantage Pro 2A gauge performs the worst partly because its location is different from the other three. One should be very careful when making such a conclusion. Maybe some analysis can be added to examine the location impact for the Davis Vantage Pro 2A gauge. For example, during winter season when precipitation is more large-scale and homogeneous, is the measurement more consistent with the other three than during summer season when precipitation is more convective and spatical heterogeneous?

There are several parts in the manuscript that the results do not have a scientific interpretation. A good physical interpretation is important to help readers understand the instruments and their measurements. Below I list some areas that I think a discussion or explanation is needed:

·        Line 355-357: the Stratus has a much lower error than the two Davis gauges [what are the errors of the Davis gauges come from?]

·        Lien 383-385: error for the Stratus gauge increases above 10mm [why?]

·        Line 391-393: the situation is different when comparing the entire period [why?]

·        Line 401-403: Stratus captures all precipitation events Hellmann recorded, but Hellmann only captures 91.6% of what Stratus recorded [is there any explanation? The inlet area of Hellmann is bigger so that evaporation is stronger? Then this will be the error source of Hellmann, which may impact the conclusion as it is treated as the “truth”]

·        Line 435-437: “automatic amateur rain gauges indicated different precipitation sums and a smaller number of days with precipitation compared to the Hellmann rain gauge.” [why?]

Detailed comments:

Title: If the analysis is treating Hellmann rain gauge as the truth and compare the two Davis gauges with the truth, the title should be something like "evaluation of selected amateur rain gauges with Hellmann rain gauge measurements".

Introduction: as suggested above, it will be good to add the possible error and uncertainty sources for each instruments. For example, as a manual instruments, how are Stratus and Hellmann gauges read by a person? Are they read by the same person at the same time? The use of “amateur” gauges looks interesting, are they (not only the two gauges in this study) operated mostly by non-experts? Is there any possible errors associated to it and any way to ensure the data quality? These would be useful information for readers who are interested in it.

The paragraph “Precipitation that …” is better moved to the first paragraph of the introduction.

Table 1: It will be good to add some more information, such as measurement frequency, price, as the paper describes analysis on these factors.

Table 2: delete "Device/".  also, change the month to either "Mon, Feb" or "01, 02". People outside Europe may not familiar with the Roman numbers.

Table 4: [MY] should be [HY]?

Round 2

Reviewer 1 Report

Accept in current form, Satisfied with revisions. 

Reviewer 2 Report

The authors addressed most of my comments. However, the revised manuscript still fails to address one of my major concerns. I understand that the authors want to emphasize that “ higher price of amateur rain gauges does not mean higher quality” as their key point, but it should really point out that this is only for one specific measured quantity (sum of the precipitation). Some gauges are higher price because they provide other things (more convenient to use, higher frequency, etc). Those related statements in abstract and conclusion should be revised otherwise they are too strong.

One other minor comment:

Line 157: when you say “it was found that in certain conditions of stronger wind systematic wind errors occur in the data.” do you have a reference to cite?
